



# Applying a new integrated mass-flux adjustment filter in rapid update cycling of convective-scale data assimilation for the COSMO-model (v5.07)

Yuefei Zeng[1], Alberto de Lozar[2], Tijana Janjic[1], and Axel Seifert[2]

[1]Meteorologisches Institut, Ludwig-Maximilians-Universität (LMU) München, Munich, Germany
[2]Deutscher Wetterdienst, Offenbach, Germany

**Correspondence:** yuefei.zeng@lmu.de

**Abstract.** A new integrated mass-flux adjustment filter is introduced, which uses the analyzed integrated mass-flux divergence field to correct the analyzed wind field. The filter has been examined by twin experiments with rapid update cycling, using an idealized setup for convective-scale radar data assimilation. It is found that the new filter slightly reduce the accuracy of background and analysis states, however, it preserves the main structure of cold pools and primary mesocyclone properties of supercells. More importantly, it considerably diminishes spurious mass-flux divergence and successfully suppresses the increase of the surface pressure tendency in the analysis. For the ensuing 3-h forecasts, the experiment that employes the filter becomes more skillful after one hour. These preliminary results show that the filter is a promising tool to alleviate the imbalance problem caused by the data assimilation.

## 1 Introduction

The performance of convective-scale data assimilation has been considerably enhanced in the last decades, greatly due to the usage of Doppler radar observations. Assimilation of radar data has been adopted in the operational mode in more and more meteorological centers (Gustafsson et al., 2018). Although weather radars provide observations in minutes, the typical update frequency at most operational centers is around one hour. Besides the high computational expense, another reason for this can be attributed to the unphysical imbalance arising from rapid update cycling. For instance, Lange and Craig (2014); Bick et al. (2016) investigated the impact of different update frequencies (5, 15, 30 and 60 min) and found that a higher update frequency results in model states which are closer to radar observations but less physically consistent (indicated by greater surface pressure tendency) and its forecast skills decay faster. In practice, a number of techniques prove to be effective to diminish the unphysical imbalance and noise produced by the data assimilation. The digital filter initialization (DFI, Lynch and Huang 1992) as well as the weak digital filter constraint (Gauthier and Thepaut, 2001) are widely used (e.g., at Meteo-France and Met Office) to damp high-frequency inertia-gravity and acoustic waves. Hydrostatic balancing of analysis increments (Rhodin et al., 2013)



is implemented in the Kilometre-scale ENsemble Data Assimilation (KENDA, Schraff et al. 2016) system for the model of COSMO (COnsortium for Small-scale MOdelling, Baldlauf et al. 2011) at the DWD to suppress the noise that is visible in the surface pressure tendency. These temporal filtering and spatial balancing methods are derived for synoptic-scale processes

and work successfully for the synoptic-scale data assimilation. Nowadays, they have been also often applied in meso-scale and convective-scale data assimilation, but their transferability onto those scales is debatable because those balance constraints may break down at high resolutions (Vetra-Carvalho et al., 2012) and high-frequency waves or unbalanced flow may be associated with meso-scale fronts or convective events. Another popular approach is the incremental analysis update (IAU, Bloom et al. 1996), which distributes the analysis increments over the assimilation window, however, it may be not adequate for rapid

cycling since the assimilation window is very short. Moreover, Zeng et al. (2017) developed an ensemble Kalman filter (EnKF) that imposes the enstrophy conservation by using a strong global constraint and tested it with a 2D shallow water model which mimics the northern hemisphere. It is shown that the enstrophy conservation reduces the noise and improves the forecast skills. For the convective scale, Ruckstuhl and Janjić (2018) tested the EnKF-based algorithm that conserves mass with the modified 1D shallow water model (Würsch and Craig, 2014) and found that the mass conservation is effective to suppress the spurious

convection. But these methods with strong constraints are very costly in computational time and essential efforts are still required to make it suitable for numerical weather prediction (NWP) models.

It is noticed that the surface pressure tendency is commonly regarded as a metric for the dynamical imbalance. Typically, the surface pressure tendency in the analysis is several times greater than that in the background (e.g., Lange and Craig 2014; Bick et al. 2016; Lange et al. 2017). Hamrud et al. (2015) introduced the surface pressure tendency as a model diagnostic

variable in a global model of the ECMWF (European Centre for Medium-Range Weather Forecasts), and the analyzed surface pressure tendency is used to adjust accordingly the analyzed horizontal wind field, which finally resulted in very small wind increments and more balanced model states. However, this approach is underdetermined because of using one variable (the surface pressure tendency) to correct two components of the horizontal wind. Therefore, some ad-hoc assumption muss be made additionally. In this work, a similar approach is developed but the increment of horizontal wind field from the analyzed

integrated mass-flux divergence and vorticity is calculated analytically. We name this new method "the integrated mass-flux adjustment filter" and examine it in a rapid update cycling of convective-scale data assimilation, using an idealized setup of the KENDA system for the COSMO model, with the data assimilation scheme of the Local Ensemble Transform Kalman Filter (LETKF, Hunt et al. 2007).

The paper is organized as follows. Section 2 describes the integrated mass-flux adjustment filter in detail. Section 3 provides a

short introduction to the idealized setup of the COSMO-KENDA system and experimental design. Section 4 gives experimental results and the last section is devoted to the conclusion and outlook.





## 2 The integrated mass-flux adjustment filter

When observations are assimilated to correct the model state, shocks can occur, which produces very large surface pressure tendency and fast waves. This can be especially remarkable for the LETKF since it is a local method and does not respect spatial derivatives.

The atmospheric surface pressure is dominated by the hydrostatic pressure:

$$p_s^h \approx \int\limits_0^{Z_0} \rho(z)gdz, \tag{1}$$

where the superscript "$h$" is for "hydrostatic", $g$ is the acceleration of gravity, $\rho$ is the air density, $z$ is the height variable and $Z_0$ is the height of the model top.

The surface pressure tendency due to changes in the hydrostatic pressure, denoted by $\frac{dp_s^h}{dt}$ can be computed using mass conservation:

$$\frac{dp_s^h}{dt} = g\int\limits_0^{Z_0} \frac{\partial \rho(z)}{\partial t}dz = -g\int\limits_0^{Z_0} \nabla_H(\rho(z)\vec{u}(z))dz = -g\nabla_H \cdot \int\limits_0^{Z_0} \rho(z)\vec{u}(z)dz = -g\nabla_H \vec{M}, \tag{2}$$

where $\vec{u} := (u,v)$ is the horizontal wind, $\vec{M} := \int_0^{Z_0} \rho(z)\vec{u}(z)dz$ is the integrated mass flux and the subscript "$H$" is for "horizontal". It is clear that surface pressure variations are determined by the divergence of the integrated mass-flux. In general, the full surface pressure tendency has a second contribution that originates from the dynamic pressure. This dynamic contribution can be larger than the hydrostatic one when large accelerations are present, like in convective regions.

Eq. 2 shows that an assimilation method that artificially increases horizontal derivatives (e.g., the LETKF) is bound to produce large surface pressure tendencies and gravity waves that might degrade the forecast. In order to mitigate this problem, Hamrud et al. (2015) used analysis increments of the surface pressure tendency as proxy for analysis increments in integrated mass-flux divergence to adjust the analysis increments for the horizontal wind.

The approach of Hamrud et al. (2015) effectively reduces the imbalance in the hydrostatic global model, but it might not be appropriate for convective-scale resolving models in which the tendencies arising from the dynamic pressure are large. Besides, this method is inaccurate in the sense that one variable (the surface pressure tendency) is used to correct two components of the horizontal wind, therefore, some ad-hoc assumption (e.g., proportionality in increments) must be made additionally. In this work, a more advanced post-processing approach is proposed. In analogy to Hamrud et al. (2015), but instead of the surface pressure tendency, we use the fact that the LETKF produces a realistic integrated mass-flux divergence if this variable is directly updated. The goal is to search for a horizontal wind field whose integrated mass-flux divergence is equal to analyzed integrated mass-flux divergence. In the following, this method is explained in detail.

For the sake of brevity, we use subscript "a" and "b" to symbolize the analysis and background states, "c" to symbolize the correction term and the superscript "*" to symbolize the final analysis state. Analyses of the integrated mass-flux divergence and vorticity are denoted by $\nabla_H \vec{M}_a$ and $\nabla_H \times \vec{M}_a$, respectively. The integrated mass-flux divergence and vorticity resulting from $u_a^*$ and $v_a^*$ are denoted by $\nabla_H \vec{M}_a^{u,v*}$ and $\nabla_H \times \vec{M}_a^{u,v*}$, respectively.





We introduce $\nabla_H \vec{M}$ as a model state variable that is updated by the LETKF, in addition to wind and other standard state variables. We set

$$\nabla_H \vec{M}_a^{u,v*} = \nabla_H \vec{M}_a, \tag{3}$$

Since Eq. 3 alone is not enough to compute $\vec{M}_a^{u,v*}$, we introduce $\nabla_H \times \vec{M}$ also as a updated model state variable and set:

$$\nabla_H \times \vec{M}_a^{u,v*} = \nabla_H \times \vec{M}_a. \tag{4}$$

Using Eqs. 3 and 4, $\vec{M}_a^{u,v*}$ can be then obtained via the Helmholzt-Hodge decomposition (see Appendix A). Now the aim is to correct the horizontal wind such that the final wind has the integrated mass flux $\vec{M}_a^{u,v*}$. Notice that the filter is based on the integrated mass flux which is a 2D field, therefore, how to distribute the integrated adjustment in the vertical is an underdetermined problem, for which we define a vertical profile function $f(z)$. In this work, we choose $f(z)$ to be large at places where the analysis increments are large:

$$f(z) = |u_a(z) - u_b(z)| + |v_a(z) - v_b(z)|. \tag{5}$$

Now we introduce a correction term $\vec{u}_c$ and it holds:

$$\vec{u}_a^*(z) = \vec{u}_a(z) + \vec{u}_c f(z), \tag{6}$$

integrating both sides of Eq. 6 with $\rho_a$ leads to:

$$\vec{u}_c = \frac{\vec{M}_a^{u,v*} - \vec{M}_a}{\int_0^{Z_0} \rho_a(z) f(z) dz}, \tag{7}$$

so $\vec{u}_c$ can be computed and finally $\vec{u}_a^*$ is computed by Eq. 6.

## 3   Description of the idealized setup and experimental design

The NWP model used in this work is the COSMO model, coupled with an Efficient Modular VOlume scanning RADar Operator (EMVORADO, Zeng et al. 2014, 2016). With the one-moment bulk microphysical scheme (Lin et al., 1983; Reinhardt and Seifert, 2006), the COSMO model predicts wind ($u$, $v$, $w$), temperature $T$, pressure $p$ and mixing ratios of water vapor $q_v$, cloud water $q_c$, cloud ice $q_i$, rain $q_r$, snow $q_s$ and graupel $q_g$. The deep convection is simulated explicitly while the shallow convection is parametrized through the Tiedtke scheme (Tiedtke, 1989). Periodic boundary conditions are used.

Configurations of the idealized setup are mostly inherited from those in Zeng et al. (2020b) with slight modifications. The grid size of the domain is now $200 \times 200 \times 65$ with the horizontal resolution of 2 km. The convection is triggered by a warm bubble in an analytical profile as Weisman and Klemp (1982).

The nature run (starting at 12:00 UTC) simulates the evolution of a supercell that is described in Fig. 1 by means of the horizontal wind at the height of 1 km, the radar reflectivity composite at the elevation of $0.5°$ and the supercell detection index



**Figure 1.** Nature run: development of the supercell from 13:00 to 18:00 UTC, described by the horizontal wind (vectors, [m/s]) at the height of 1 km and the radar reflectivity composite of $0.5°$ (dotted contour lines for 30 dBZ) and the $SDI^2$ [1/s] (color shading)





of the second type (SDI$^2$, Wicker et al. 2005; Zeng et al. 2020b). The SDI$^2$ identifies the mesocyclone of a supercell, while a positive (negative) SDI$^2$ indicates a cyclonic (anticyclonic) updraft. $\left|\mathrm{SDI}^2\right| > 0.0003$ 1/s is a minimal threshold value for a supercell. A small convective system arises at 13:00 UTC and moves northeastwards. At 14:00 UTC, it splits into two cells, while the left-moving one is anticyclonic and the right-moving one is cyclonic. At 15:00 UTC, both cells grow a bit and remain stable until 18:00 UTC.

A radar network of 6 stations covering the whole domain is mimicked, and the synthetic radar observations are created by adding Gaussian noise with standard deviation of 1.0 m/s and 5.0 dBZ to the true radial wind and reflectivity data, respectively. As in many studies (e.g., Aksoy et al. 2009; Zeng et al. 2018, 2019, 2020a), to suppress spurious convection, all reflectivities lower than 5 dBZ are set to 5 dBZ and treated as no reflectivity data. Both radial wind and reflectivity (including no reflectivity) data are assimilated. The radar observations are available every 5 min and spatially superobbed to the resolution of 5 km. Since

the data assimilation is the LETKF, the localization is done in the observation space. The vertical localization radius varies with the height from 0.0075 to 0.5 in logarithm of pressure and the horizontal localization radius is set to be 8 km. A diagonal observation error covariance matrix is used.

All the prognostic model variables mentioned above are updated. 45 ensemble members are employed, whose initial states differ in the atmospheric profile and the location of warm bubble (more details can be found in Zeng et al. 2020b). Additionally,

one deterministic run is initialized by the mean state of the ensemble. To maintain sufficient ensemble spread, the relaxation-to-prior-perturbations method (RTPP, Zhang et al. 2004) is used with a relaxation factor of 0.75. The data assimilation cycling period is from 13:00 to 15:00 UTC with the update frequency of six minutes. Ensemble and deterministic forecasts with 3-h lead time are run starting at 15:00 UTC.

In this work, two twin experiments are conducted, denoted by E_VrZ_6m and E_VrZ_6m_f. The former one is run without

the integrated mass-flux adjustment filter (hereafter filter in short if not explicitly mentioned) and the latter one is run with the filter.

## 4 Experimental results

For the investigation of the performance during assimilation cycles, we mainly use the deterministic run instead of the ensemble mean since some small-scale features that we are interested in may be smoothed out in the mean. We calculate the root-

mean-square errors (RMSEs) and spreads, integrated mass-flux divergence and surface pressure tendency of the background and analysis. Additionally, temperature and moisture deviations in the sub-cloud layer (calculated as deviations of vertical average over 2 km from the horizontal mean, denoted by $\Delta T$ and $\Delta q_v$, respectively) are used to discuss the structure of cold pools (Seifert and Heus, 2013) and the SDI$^2$ is used to compare the ability to reconstruct the mesocyclone of supercells in analysis, for which the ensemble probabilities from the number of members exceeding a threshold value are computed in

a neighborhood manner (Yussouf et al., 2013). To evaluate the forecast skill, gridpoint-based ensemble probabilities for the reflectivity composite are provided and the fractions skill score (FSS, Robert and Lean 2008) is used. The FSS value varies between 0 and 1, with 1 being the perfect score.



## 4.1 Assimilation

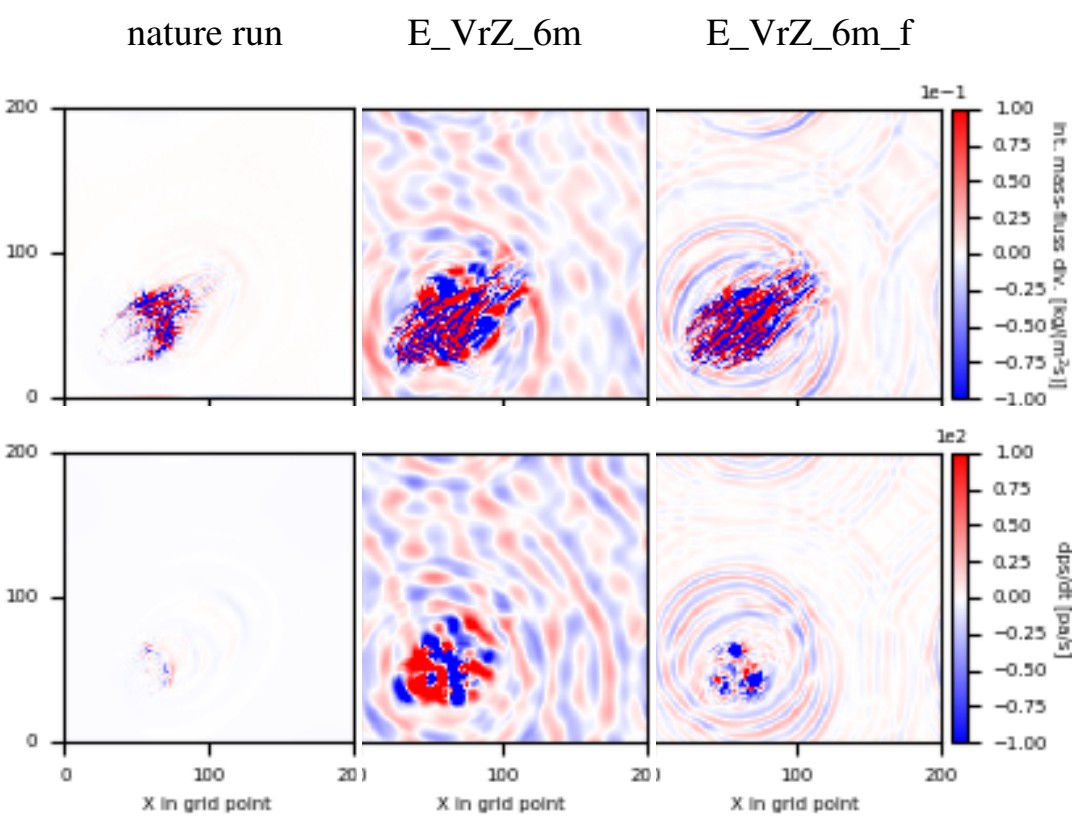

**Figure 2.** The upper row is for the integrated mass-flux divergence of nature run (left), analyses of E_VrZ_6m (middle) and E_VrZ_6m_f (right) at 15:00 UTC; the second row is for the surface pressure tendency

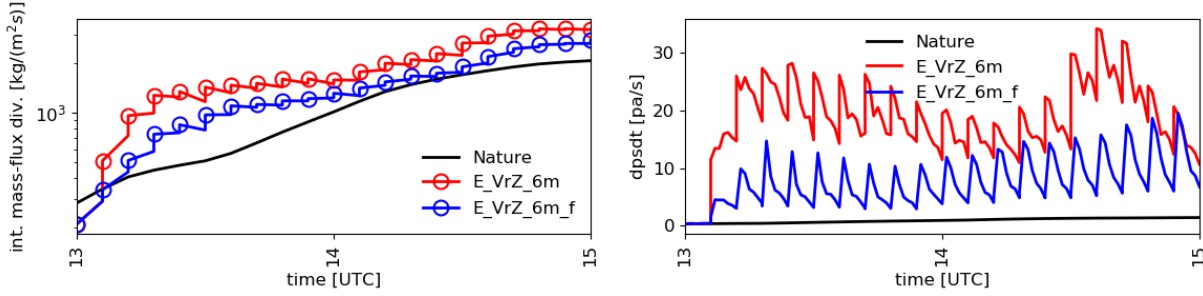

**Figure 3.** Domain integrated mass-flux divergence of analysis (indicated by circle) and background at each cycle (left) and domain-averaged surface pressure tendency at the analysis time and at each minute of model integration during the assimilation window (right), for E_VrZ_6m and E_VrZ_6m_f, compared with those of the nature run



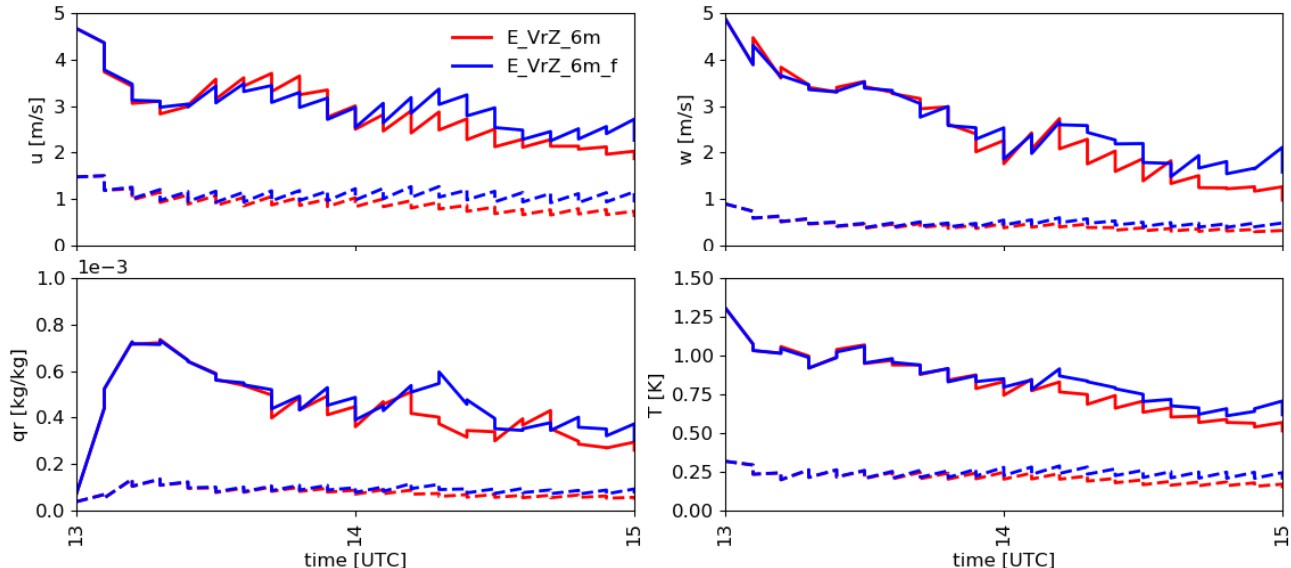

**Figure 4.** The RMSEs (solid line) and spreads (dashed line) of the deterministic background and analysis (sawtooth pattern) for $u$, $w$, $q_r$ and $T$ at each cycle in E_VrZ_6m and E_VrZ_6m_f, averaged over points at which the reflectivity in the nature run is greater than 5 dBZ.

Fig. 2 compares E_VrZ_6m and E_VrZ_6m_f with the nature run in terms of integrated mass-flux divergence and surface

pressure tendency fields of the deterministic analysis at 15:00 UTC. Both experiments produce stronger integrated mass-flux divergence fields than the nature run but E_VrZ_6m_f is considerably better than E_VrZ_6m. The similar can be seen also for the surface pressure tendency field, the gravity waves are much weaker in E_VrZ_6m_f. Apropos, it is noticed that the surface pressure tendency correlates well with the integrated mass flux divergence only outside the convective regions but not inside, which justifies using the updates of the integrated mass flux instead of the surface pressure tendency in our filter.

Using the same metrics, Fig. 3 further compares both experiments for analysis and background at each cycle. It is evident that both results in analyses and backgrounds with larger integrated mass-flux divergence than the nature run but E_VrZ_6m_f is considerably closer. The surface pressure tendency significantly increases at the analysis step and rapidly decays in the model integration time but does not reach the level of the nature run. E_VrZ_6m_f generates much lower surface pressure tendencies than E_VrZ_6m for both analysis and background. Overall, it is concluded that the filter effectively reduces the gravity wave

noise caused by the data assimilation and thus improves the balance of model states.

Fig. 4 gives the RMSEs and spreads of the deterministic background and analysis for $u$, $w$, $q_r$ and $T$ at each cycle. Both E_VrZ_6m and E_VrZ_6m_f exhibit the effectiveness of data assimilation based on decreasing RMSEs, but in general, the former one is associated with slightly smaller RMSEs and spreads for $u$. This holds also for the other variables. Therefore, the application of the filter causes some loss of analysis accuracy. It is noted that this problem is also often seen by the other

filtering methods (e.g., the DFI in Ancell 2012).

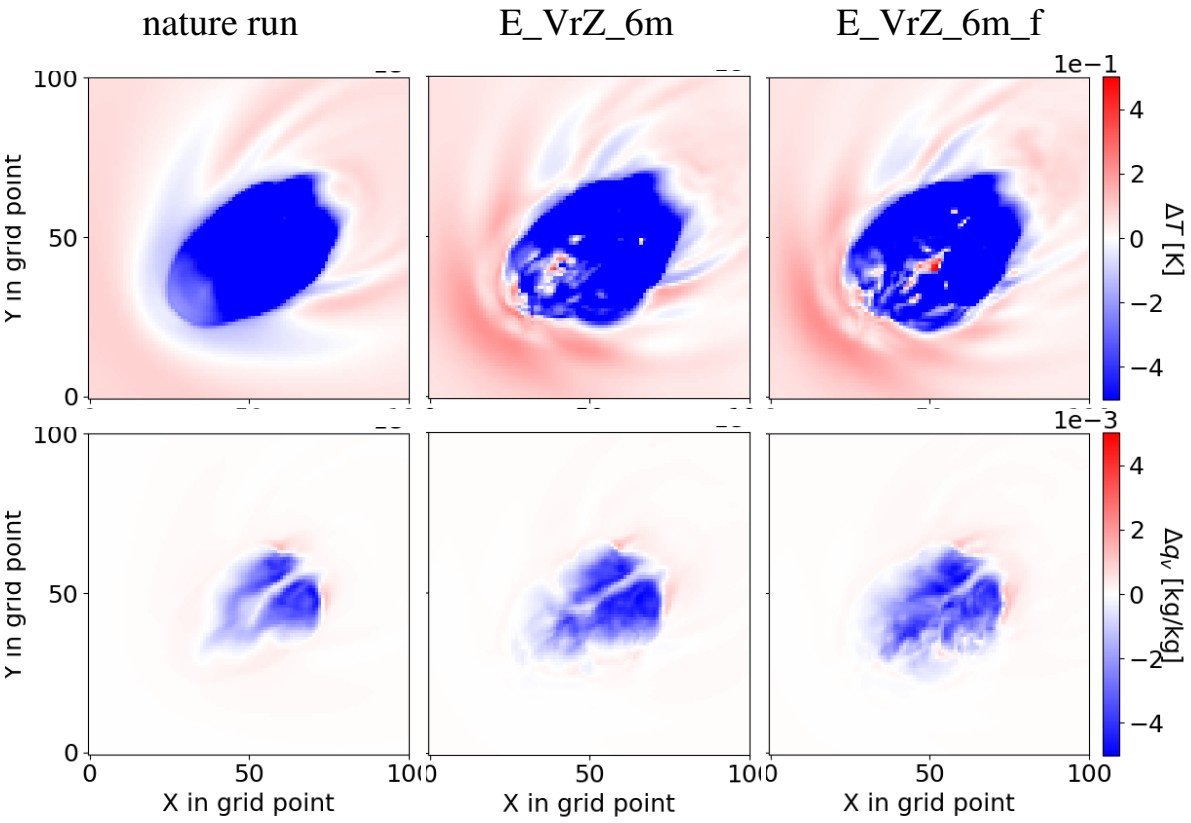

**Figure 5.** Temperature (upper) and moisture (lower) deviations in the sub-cloud layer for the nature run (left) and analyses of E_VrZ_6m (middle) and E_VrZ_6m_f (right) at 15:00 UTC

As known, the presence of cold pool (caused by evaporation of rain and downdrafts) could be important for the development of deep convection (Böing et al., 2012). Fig. 5 compares E_VrZ_6m and E_VrZ_6m_f with the nature run in terms of sub-cloud layer deviations $\Delta T$ and $\Delta q_v$ at 15:00 UTC. The cold pool in E_VrZ_6m is fairly comparable to the truth, although it is generally a bit drier and it is warmer to different extents in the rear and at some spots on the inside. The cold pool in

E_VrZ_6m_f is very similar to that in E_VrZ_6m. Therefore, the cooling/moistening of the sub-cloud layer is well reproduced in both experiments and the filter does not deteriorate the basic structure of the cold pool.

In Fig. 6, the $SDI^2$ of the deterministic analysis at 15:00 UTC in E_VrZ_6m (left) and E_VrZ_6m_f (right) is given. It is noticed that both experiments capture well the major events of mesocyclones although some spurious convection exists. Also for the ensemble probabilities, good performance of both experiments are evident, e.g., very high $SDI^2$ probabilities (till 100%)

can be seen at locations where $SDI^2 > 0.0003$ 1/s is present in the nature run, which indicates that the filter preserves primary characteristics and structures of mesocyclones.





**Figure 6.** SDI$^2$ of analysis of the deterministic run (upper) and ensemble (lower) for E_VrZ_6m (left) and E_VrZ_6m_f (right) at 15:00 UTC. SDI$^2 \geq$ 0.0003 1/s in the nature run is denoted with contour dashed lines. For the ensemble, probabilities exceeding the threshold value 0.0003 1/s are calculated using neighborhood of boxlength 8 km.

## 4.2 Forecasting

Fig. 7 provides the surface pressure tendency during the 3-h forecast of E_VrZ_6m and E_VrZ_6m_f for the deterministic run. Compared to E_VrZ_6m, the surface pressure tendency of E_VrZ_6m_f is much lower at the initial time but it increases

somehow in the first few minutes, which indicates some spin-up time after the filter, and then decays rapidly to a steady level



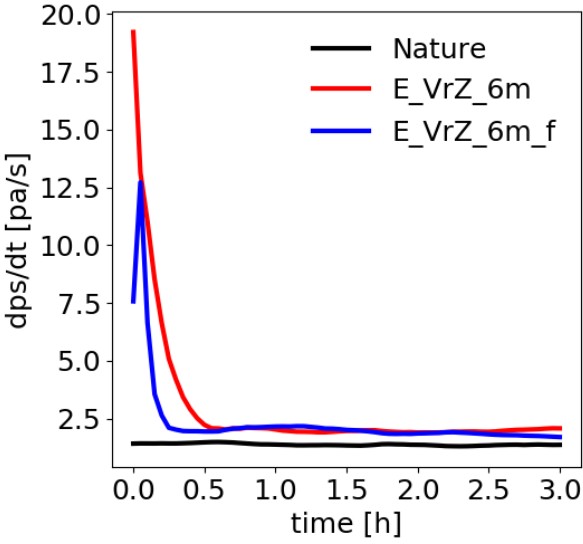

**Figure 7.** Domain-averaged surface pressure tendency at each minute during the 3-h deterministic forecast starting at 15:00 UTC for the nature run, E_VrZ_6m and E_VrZ_6m_f

close to the nature run in 15 min. Comparatively, the surface pressure tendency of E_VrZ_6m approaches to the nature run a bit later in 30 min.

Fig. 8 presents deterministic and ensemble forecasts by the reflectivity composite at the elevation $0.5°$. For the deterministic forecast, both E_VrZ_6m and E_VrZ_6m_f well represent the storms but with some spurious convection in the rear; For 1-h lead time, the right-moving cell is slightly better captured by E_VrZ_6m, but for 2-h and 3-h lead times, both cells especially the right-moving one is slightly better captured by E_VrZ_6m_f. It is noticed that a considerable amount of spurious convection arise at 3-h lead time in both experiments. For the ensemble forecast as shown in Fig. 9, both experiments are comparable at the initial time and 1-h lead time, but for 2-h and 3-h lead times, E_VrZ_6m_f outperforms E_VrZ_6m as the former one produces less spurious convection. Those results are also confirmed by the FSS in Fig. 10. For the deterministic forecast, E_VrZ_6m is better than E_VrZ_6m_f until 2-h lead time and then E_VrZ_6m_f is better; For the ensemble forecast, E_VrZ_6m_f becomes superior already at 1-h lead time.

## 5 Summary and outlook

A new integrated mass-flux adjustment filter that uses the analyzed integrated mass-flux divergence field to correct the analyzed wind field has been introduced in this work. The filter has been examined by twin experiments with rapid update cycling (i.e., six minutes), using an idealized setup for convective-scale radar data assimilation. It is found that the new filter slightly reduces the accuracy of background and analysis states, however, it preserves the main structure of cold pools (described by



**Figure 8.** Deterministic forecasts (starting at 15:00 UTC) of E_VrZ_6m (middle) and E_VrZ_6m_f (right), based on the reflectivity composite at the elevation $0.5°$. The forecasts at the initial time, 1h, 2 h and 3 h are presented from top to bottom. The left column is the nature run.

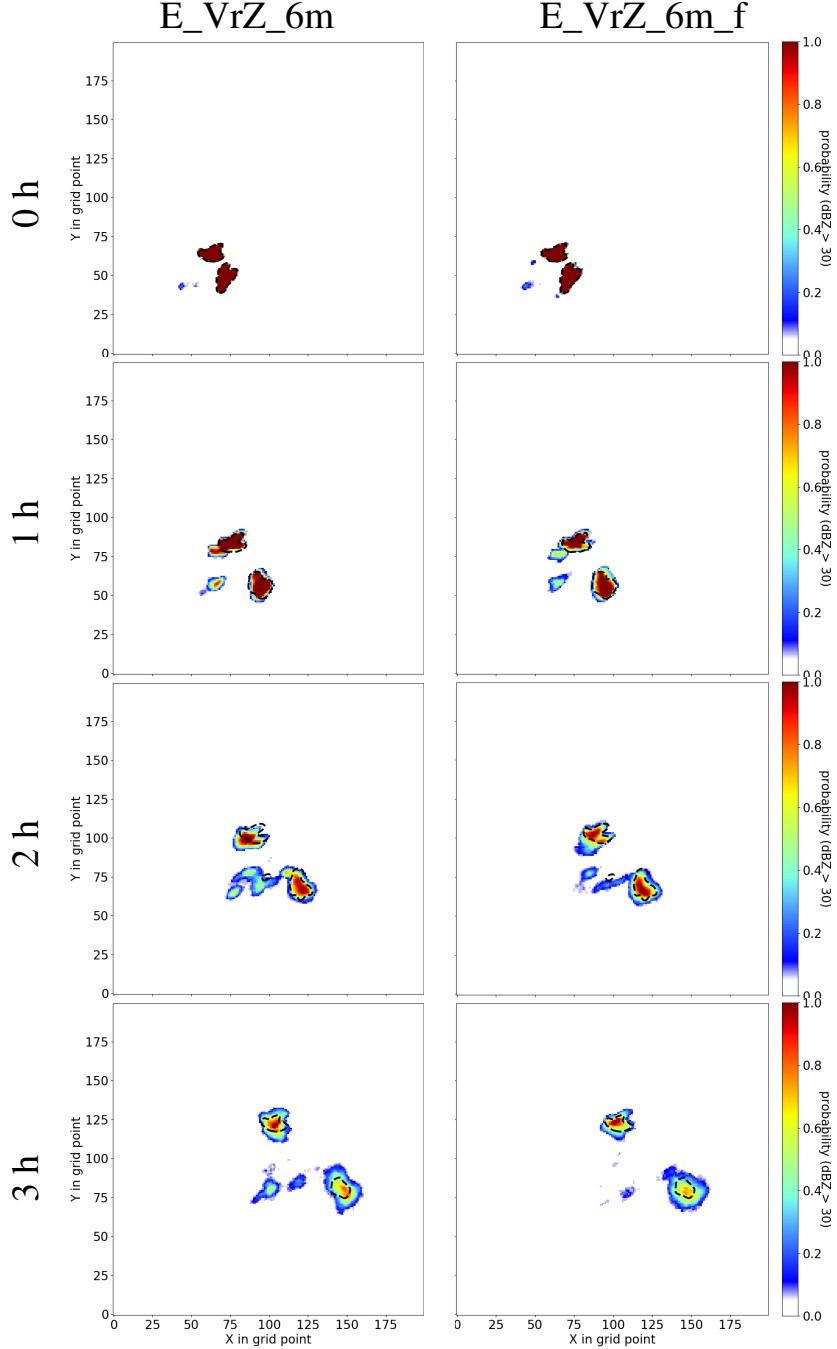

**Figure 9.** Ensemble forecasts (starting at 15:00 UTC) of E_VrZ_6m (left) and E_VrZ_6m_f (right), based on gridpoint-based probabilities of ensemble members with reflectivity composite at elevation $0.5°$ exceeding the threshold value 30 dBZ. Reflectivity $\geq 30$ dBZ in the nature run is denoted with contour dashed lines. The forecasts at the initial time, 1h, 2 h and 3 h are presented from top to bottom.





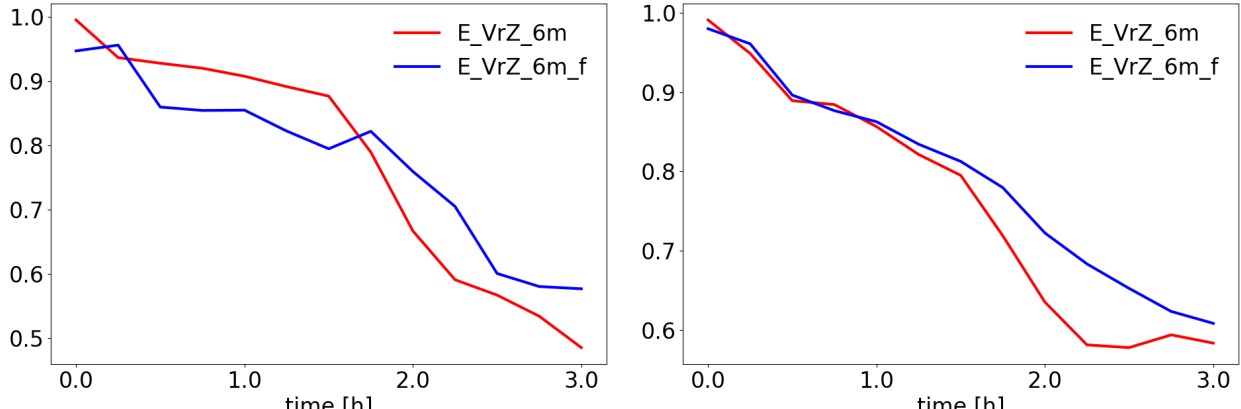

**Figure 10.** The FSS values of the deterministic (left) and ensemble forecasts (right) starting at 15:00 UTC in E_VrZ_6m and E_VrZ_6m_f, based on reflectivity composite at $0.5°$ with the threshold value 30 dBZ and boxlengths 16 km, as a function of the forecast lead time up to 3 hours (with 15 min outputs). The FSS values of ensemble forecasts are averaged over ensemble members.

temperature and moisture deviations in the sub-cloud layer) and primary mesocyclone properties of supercells (described the supercell section index). More importantly, it considerably diminishes spurious integrated mass-flux divergence and successfully suppresses the increase of the surface pressure tendency in analysis. For the ensuing 3-h forecasts, the one that employes

the filter becomes more skillful after one hour. Overall, the filter is a promising tool to alleviate the imbalance problem usually caused by the data assimilation.

With respect to the further development, the filter is currently based on the 2D integrated mass-flux field and the adjustments in the vertical are distributed by a predefined function, therefore, it is reasonable to extend the filter to 3D, which means the wind is corrected by analyzed mass-flux at each level individually. Although the filter has been tested here for convective-scale

data assimilation with rapid cycling, it can be also useful for lower update frequencies and the synoptic-scale data assimilation. This is worth exploring in the future, as well as its applicability in the non-idealized data assimilation.

**Appendix A**

The Helmholtz-Hodge decomposition says that any vector field tangent to the surface of the sphere can be written as the sum

$$\vec{f} = \nabla_H \phi + \nabla_H \times \psi, \tag{A1}$$

where $\phi$ and $\psi$ are scalar potential functions. In this work, $\vec{f} = \vec{M}_a^{u,v*}$.

Because the divergence of a curl is zero, we obtain the Poisson equation:

$$\nabla_H \vec{M}_a^{u,v*} = \nabla_H \cdot \nabla_H \phi = \nabla_H^2 \phi. \tag{A2}$$

Due to Eq. 3, it holds

$$\nabla_H^2 \phi = \nabla_H \vec{M}_a \tag{A3}$$





Solving the Poisson equation A3, we obtain $\phi$.

Because the vorticity of a gradient field is zero, we obtain the Poisson equation:

$$\nabla_H^2 \psi = \nabla_H \times \vec{M}_a \tag{A4}$$

in an analogous way. Solving Eq. A4, we obtain $\psi$.

Finally, substituting $\phi$ and $\psi$ in Eq. A1, we obtain $\vec{M}_a^{u,v*}$.

*Code and data availability.*

All the data upon which this research is based are available through personal communication with the authors. The code of the new filter is written in Python and it is available at the following link: https://doi.org/10.5281/zenodo.4023987. Access to the source code of the COSMO model is restricted to COSMO licenses. A free license can be obtained for research if following the procedure described at http://www.cosmo-model.org/cotent/consortium/licencing.htm.

*Author contributions.*

Y. Zeng conducted the experiments and wrote the first draft of the paper. A. de Lozar provided the idea of the integrated mass-flux filter and implemented it. T. Janjic and A. Seifert contributed to the conceptual design of the research project and analysis of the results. All authors contributed to the writing of the text and defining the structure of the paper.

*Competing interests.*

The authors declare that they have no conflict of interest.

*Acknowledgements.* Thanks are given to the DFG (Deutsche Forschungsgemeinschaft) Priority Program 2115:PROM through the project JA 1077/5-1. The work of T. Janjic is supported through DFG Heisenberg Programm JA 1077/4-1. The work of A. de Lozar is supported by the Innovation Programm for applied Research and Development (IAFE) of Deutscher Wetterdienst in the framework of the SINFONY project. Thanks are also given to Christian A. Welzbacher from the DWD and Leonhard Scheck from the Hans Ertel Centre for Weather Research
(Weissmann et al., 2014; Simmer et al., 2016) at the LMU for technical supports.

*Financial support.*

This research has been supported by the DFG.





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
