# Peer review of "Applying a new integrated mass-flux adjustment filter in rapid update cycling of convective-scale data assimilation for the COSMO-model (v5.07)"

_Geoscientific Model Development, 2020_

## Referee Comment (RC1) · Anonymous Referee #1 · 22 Dec 2020

The paper introduced a new integrated mass-flux adjustment filter in Ensemble Kalman Filter (EnKF) to correct the analyzed wind field and suppress the unphysical increase of the surface pressure tendency in the analysis. An idealized supercell storm was used to examine the performance of the new filter. The root-mean-square error, ensemble spread, cool pool, surface pressure tendency, and supercell detection index were investigated. The results show that the new filter slightly degrades the analysis accuracy, which is still acceptable, but this filter alleviates the imbalance problem caused by the data assimilation. The forecast skill in terms of fractions skill scores (FSSs) of reflectivity composite and the number of spurious convection is improved after using the new filter. This paper is interesting and well-written. I recommend that the paper should be accepted with Minor revisions and I include my few comments below.

Specific comments L5-6: Readers who are not familiar with dynamic problems associated with data assimilation may be confused with the words: "suppress the increase of the surface pressure tendency in the analysis". Please spend a bit more words on why the increase of the surface pressure tendency in the analysis should be suppressed.

L63-66: Why exclude the vertical mass flux?

L76-78: How to understand the words: "a realistic integrated mass-flux divergence if this variable is directly updated?" Do authors mean that using the cross-variable covariance between observations (e.g., HX of Vr and HX of Z) and the integrated mass-flux divergence to update? If so, please directly tell readers how to update the integrated mass-flux divergence and think about whether the word "realistic" is suitable here, because an accurate analysis depends on the accuracy of covariance which is not also reliable in EnKF especially in the first few cycles.

L91-92: Please tell the physical meaning of this function. Why design the function in the form of Eq. (5).

L105: Please briefly list some key points of configurations in Zeng et al (2020b)

L115: If possible, add a plot of radar locations or list the radar locations. I am not sure whether radars observed the entire storm, especially at low levels. Without low-level airflow information, the analysis of integration mass-flux divergence may not be accurate as expected.

L124: Environment errors were introduced? A brief description of the difference between profiles will be appreciated.

L126: Why is 0.75?

[Figure]

Figure 3: It seems that the imbalance mass flux mainly affects the first few cycles. The amplitudes of surface pressure tendency in E_VrZ_6m are not much larger than those in E_VrZ_6m_f after the first few cycles, except for those after 14:30 UTC. If stop using the mass-flux filter after the first several cycles, what will happen? In addition, please adjust the position of the legend in Figure 3b (the right one).

Figure 4: The loss of accuracy is OK, but it is better to concern the relatively rapid increase of forecast error in u just after 14 UTC. Reducing mass-flux error does not certainly ensure a lower forecast error? Additionally, in some analyses after 14 UTC, the RMSE of qr becomes larger after analysis. It seems that the cross-variable error covariance is not so reliable after using the mass-flux filter. A bit more discussion on the potential negative impact of using the new filter will be helpful for others who would like to adopt the filter.

Figure 5: It is a good result, but what is the physical relationship between the mass-flux filter and this better cold pool? Is it valid in most cases or is case dependent?

L180-181: Please directly point out what is better. The areas of spurious convection are smaller? The environment perturbation may also introduce spurious convections. How to extract the contribution of the new mass-flux filter from the final forecast results?

---

## Referee Comment (RC2) · Anonymous Referee #2 · 12 Jan 2021

This manuscript proposed a new integrated mass-flux adjustment filter. For the convective-scale data assimilation, data assimilation cycles from a twin experiment showed that the integrated mass-flux adjustment preserved the main structure of cold pools and primary mesocyclone properties of supercells, although it degraded the priors and posteriors. The 3-h free forecast showed that the integrated mass-flux adjustment obtained more skillful forecasts after one hour and alleviated the imbalance caused by data assimilation, although the surface pressure tendency showed a spin-up feature. The integrated mass-flux adjustment for the LETKF is applied for rapid update

cycling of convective-scale data assimilation in this study, but it can also be applied for synoptic-scale data assimilation. Imbalance caused by intermittent data assimilation is an essential problem, especially for applications favorable balanced atmospheric states. The manuscript is scientifically sound and well written. My recommendation is between minor and major. Please see detailed comments as below.

1. l28-30, this statement about IAU is unclear. There are four-dimensional IAU (4DIAU) that takes into account temporal variations of increments and has advantages over the commonly used 3DIAU (Lei and Whitaker 2016). Thus the IAU could be suitable for rapid cycling with short data assimilation windows. Moreover, a recent study showed that with more frequent updates, i.e., short data assimilation windows, the imbalance caused by data assimilation is reduced, while the 3DIAU/4DIAU are still helpful to reduce the imbalance but with smaller impacts (He et al. 2020).

2. l92-93, how the integrated mass flux method be sensitive to the choice of f(z)? Any validation for the choice of f(z)?

3. l120-121, it would be helpful to give the function of vertical localization length scales.Âă

4. l133-134, how to get the priors and posteriors of the deterministic forecast for the assimilation cycles? This question also applies to the plot contents of Figures 3-6.

5. l148-149, is this the opposite? The correlations between integrated mass flux divergence and surface pressure tendency are mainly "inside" the convective regions?

6. l157-159, it would be helpful to provide some explanations for the degradation of errors caused by the integrated mass flux divergence. Intuitively, by adjusting the integrated mass flux, a more balanced analysis could be obtained, which is preferable for improved forecasts. Could this intuitive hypothesis be true for large scale applications? Since E_VrZ_6m_f has larger errors and spread than E_VrZ_6m, especially for later times, E_VrZ_6m_f might have smaller increments than E_VrZ_6m. The smaller

increments might not be large enough to correct the prior errors, although it is better balanced?Âă

7. l175, why the spin-up shows in E_VrZ_6m_f but not in E_VrZ_6m? Since the spin-up show in E_VrZ_6m, does it mean that the adjustment based on integrated mass flux is too much?Âă

8. l178-86, the results of forecasting are based one forecast launched at 15:00 UTC. To draw statistically significant conclusions, more than one forecast is preferred. Can the forecast be launched at different hours, so that more general conclusions can be obtained?

References: H. He, L. Lei, J. S. Whitaker, and Z.-M. Tan, 2020: Impacts of Assimilation Frequency on Ensemble Kalman Filter Data Assimilation and Imbalances. J. Adv. Model. Earth Syst., 12, e2020MS002187, doi: https://doi.org/10.1029/2020MS002187.

Lei, L., and J. S. Whitaker, 2016: A four-dimensional incremental analysis update for the ensemble Kalman filter. Mon. Wea. Rev., 144, 2605-2621. doi: http://dx.doi.org/10.1175/MWR-D-15-0246.1.
* * *

---

## Author Comment (AC1) · 27 Jan 2021

**To Reviewer 1**

The paper introduced a new integrated mass-flux adjustment filter in Ensemble Kalman Filter (EnKF) to correct the analyzed wind field and suppress the unphysical increase of the surface pressure tendency in the analysis. An idealized supercell storm was used to examine the performance of the new filter. The root-mean-square error, ensemble spread, cool pool, surface pressure tendency, and supercell detection index

were investigated. The results show that the new filter slightly degrades the analysis accuracy, which is still acceptable, but this filter alleviates the imbalance problem caused by the data assimilation. The forecast skill in terms of fractions skill scores (FSSs) of reflectiv- ity composite and the number of spurious convection is improved after using the new filter. This paper is interesting and well-written. I recommend that the paper should be accepted with Minor revisions and I include my few comments below.

**Answer:** Thank you very much for your kind acknowledgment of our work.

Specific comments L5-6: Readers who are not familiar with dynamic problems associated with data assimilation may be confused with the words: "suppress the increase of the surface pressure tendency in the analysis". Please spend a bit more words on why the increase of the surface pressure tendency in the analysis should be suppressed.

**Answer:** We rephrased the sentence as "it considerably diminishes spurious mass-flux divergence as well as the high the surface pressure tendency and thus results in more dynamically balanced analysis states".

L63-66: Why exclude the vertical mass flux?

**Answer:** We are not sure if we understand the question exactly. The use of the integrated mass flux for the filter is valid due to the analogy of the integrated mass-flux divergence to the surface pressure tendency as shown in Eq. 2, and the vertical mass flux does not appear in Eq. 2. Furthermore, as mentioned in the outlook, we will extend the filter to the vertical, i.e., correcting the wind field by analyzed mass-flux from level

to level, which may be more accurate and more balanced but computationally more expensive.

L76-78: How to understand the words: "a realistic integrated mass-flux divergence if this variable is directly updated?" Do authors mean that using the cross-variable covariance between observations (e.g., HX of Vr and HX of Z) and the integrated mass-flux divergence to update? If so, please directly tell readers how to update the integrated mass-flux divergence and think about whether the word "realistic" is suitable here, because an accurate analysis depends on the accuracy of covariance which is not also reliable in EnKF especially in the first few cycles.

**Answer:** Note that the integrated mass-flux divergence filter does not correct the wind field via the cross covariance and it is a post-processing method. Eqs. 1-3 are equations of LETKF Hunt et al. (2007). It can be seen that, adding variables such as integrated mass-flux divergence which do not go to the observation forward operator does not change $\tilde{\mathbf{P}}_k^a$ and does not change $\overline{\mathbf{w}}_k^a$, therefore, it has no influence on the other variables of $\overline{\mathbf{x}}_k^a$. But on the other hand, the changes in variables like $u$ that go to the observation forward operator will change the integrated mass-flux divergence through cross correlations. Moreover, the wording of "realistic" is correct because the integrated mass-flux divergence derived from the wind field of the analysis is usually significantly larger than that derived from background. But if treating integrated mass-flux divergence as a variable and using the LETKF to update it, the resulted integrated mass-flux divergence of the analysis is slightly larger than that of the background as seen in Fig. 5. We rephrased as "we use the fact that we can soundly estimate integrated mass-flux divergence via LETKF directly through cross correlations".

$$\overline{\mathbf{x}}_k^a = \overline{\mathbf{x}}_k^f + \mathbf{X}_k^f \overline{\mathbf{w}}_k^a \tag{1}$$
$$\overline{\mathbf{w}}_k^a = \tilde{\mathbf{P}}_k^a \mathbf{Y}_k^f \mathbf{R}_k^{-1} \left( \mathbf{y}_k^o - \overline{\mathbf{y}}_k^f \right) \tag{2}$$

$$\tilde{\mathbf{P}}_k^a = \left[ (N-1)\mathbf{I} + \mathbf{Y}_k^{f^T} \mathbf{R}_k^{-1} \mathbf{Y}_k^f \right]^{-1} \tag{3}$$

L91-92: Please tell the physical meaning of this function. Why design the function in the form of Eq. (5).

**Answer:** The function $f$ should distribute the integrated adjustment over the column to correct the wind field. We assume the corrections should be larger at places where the analysis increments of the wind field are larger. We rephrased the sentence.

L105: Please briefly list some key points of configurations in Zeng et al (2020b)

**Answer:** We added "the analytical profile is defined by two parameters $u\_infty$ and $q_v\_$max. $u\_infty$ is the upper wind in the troposphere, which determines the entire wind profile and scales the wind shear, and $q_v\_$max determines the humidity profile and a higher $q_v\_$max results in stronger instability of the atmosphere".

L115: If possible, add a plot of radar locations or list the radar locations. I am not sure whether radars observed the entire storm, especially at low levels. Without low level airflow information, the analysis of integration mass-flux divergence may not be accurate as expected.

**Answer:** The plot of radar locations is added (see Figure 1). As seen, it covers the propagation path of storms within the study period.

L124: Environment errors were introduced? A brief description of the difference

between profiles will be appreciated.

**Answer:** Done

L126: Why is 0.75?

**Answer:** The coefficient is tuned to 0.75 for KENDA system (Schrafff et. al 2016). We rephrased it.

Figure 3: It seems that the imbalance mass flux mainly affects the first few cycles. The amplitudes of surface pressure tendency in E_VrZ_6m are not much larger than those in E_VrZ_6m_f after the first few cycles, except for those after 14:30 UTC. If stop using the mass-flux filter after the first several cycles, what will happen? In addition, please adjust the position of the legend in Figure 3b (the right one).

**Answer:** High frequency convective-scale data assimilation continually accumulates noises and imbalances through cycling, stoping using the filter will certainly lead to larger increase in surface pressure tendency. We changed the layout of the legend.

Figure 4: The loss of accuracy is OK, but it is better to concern the relatively rapid increase of forecast error in u just after 14 UTC. Reducing mass-flux error does not certainly ensure a lower forecast error? Additionally, in some analyses after 14 UTC, the RMSE of qr becomes larger after analysis. It seems that the cross-variable error covariance is not so reliable after using the mass-flux filter. A bit more discussion on the potential negative impact of using the new filter will be helpful for others who would like to adopt the filter.

**Answer:** The filter is a post-processing approach that does not consider the accuracy of model states while reducing the imbalance. An idea that takes those both regards into account is to impose the filter as a weak constraint to the cost function of EnKF algorithms. We added texts for this in the outlook.

Figure 5: It is a good result, but what is the physical relationship between the mass-flux filter and this better cold pool? Is it valid in most cases or is case dependent?

**Answer:** Strongly rotating updraft (i.e., mesocyclone) and cold pool are important features of supercells. The application of the filter could possibly deteriorate those features as it causes some accuracy loss to model states. However, Figure 6 shows that the filter generates a comparable cold pool as without using the filter.

L180-181: Please directly point out what is better. The areas of spurious convection are smaller? The environment perturbation may also introduce spurious convections. How to extract the contribution of the new mass-flux filter from the final forecast results?

**Answer:** We added "Results indicate that the application of the filter reduces the dynamical imbalance of analyses, which slows down the error growth of model states in free forecasts and thus improves the forecast skills. This is in line with Zeng and Janjic (2016); Zeng et al. (2017)".

[Figure]

**Fig. 1.** Distribution of radar network (six radars), expressed by the range of PPI (Plan Position Indicator) scan at the elevation $0.5^{\circ}$

---

## Author Comment (AC2) · 27 Jan 2021

**To Reviewer 2**

**Major Comments**

This manuscript proposed a new integrated mass-flux adjustment filter. For the convective-scale data assimilation, data assimilation cycles from a twin experiment showed that the integrated mass-flux adjustment preserved the main structure of cold

pools and primary mesocyclone properties of supercells, although it degraded the priors and posteriors. The 3-h free forecast showed that the integrated mass-flux adjustment obtained more skillful forecasts after one hour and alleviated the imbalance caused by data assimilation, although the surface pressure tendency showed a spin-up feature. The integrated mass-flux adjustment for the LETKF is applied for rapid update cycling of convective-scale data assimilation in this study, but it can also be applied for synoptic-scale data assimilation. Imbalance caused by intermittent data assimilation is an essential problem, especially for applications favorable balanced atmospheric states. The manuscript is scientifically sound and well written. My recommendation is between minor and major.

**Answer:** Thank you very much for your kind acknowledgment of our work.

1. l28-30, this statement about IAU is unclear. There are four-dimensional IAU (4DIAU) that takes into account temporal variations of increments and has advantages over the commonly used 3DIAU (Lei and Whitaker 2016). Thus the IAU could be suitable for rapid cycling with short data assimilation windows. Moreover, a recent study showed that with more frequent updates, i.e., short data assimilation windows, the imbalance caused by data assimilation is reduced, while the 3DIAU/4DIAU are still helpful to reduce the imbalance but with smaller impacts (He et al. 2020).

**Answer:** The IAU has been used in practice for convective-scale data assimilation, but the update frequencies are usually not shorter than one hour. The performance of IAU for the ultra-rapid update cycle such as 6 min in this work has not been examined so far to our knowledge. Results of He et al. 2020 are based on large-scale data assimilation with update frequencies from 12 hours to 1 hour. It is very different in spatial and temporal scales of our work. Therefore, their conclusion may not hold for convective-scale data assimilation. Actually as shown in Bick et al. 2016, the rapid

updates (from 1 hour to 5 min) keep imbalance at high levels within cycles. The same statement can be also found in Pierre Brousseau et al. 2008. We rephrased the text.

Bick, T., Simmer, C., Trömel, S., Wapler, K., Stephan, K., Blahak, U., Zeng, Y., and Potthast, R.: Assimilation of 3D-Radar Reflectivities with an Ensemble Kalman Filter on the Convective Scale, Quart. J. Roy. Meteor. Soc., 142, 1490-1504, 2016.

Pierre BROUSSEAU, Francois BOUTTIER, Gwena|elle HELLO, Yann SEITY, Claude-FISCHER, Loik BERRE, Thibaut MONTMERLE, Ludovic AUGER, Sylvie MALARDEL: A prototype convective-scale data assimilation system for operation : the Arome-RUC, HIRLAM Technical Report No. 68, 2008

2. l92-93, how the integrated mass flux method be sensitive to the choice of f(z)? Any validation for the choice of f(z)?

**Answer:** The function $f$ should distribute the integrated adjustment over the column to correct the wind field. We assume the corrections should be larger at places where the analysis increments of the wind field are larger. The idea is similar to Hamrud et al. 2015, who used the analysis spread of the wind field instead of the analysis increment. We rephrased the sentence.

3. l120-121, it would be helpful to give the function of vertical localization length scales.

**Answer:** We added "i.e., the weights assigned to observations are scaled by the the 5-th order Gaspari-Cohn function (Gaspari and Cohn, 1999), which depends on the vertical and horizontal distances of observations to the analysis grid point".

4. l133-134, how to get the priors and posteriors of the deterministic forecast for the assimilation cycles? This question also applies to the plot contents of Figures 3-6.

**Answer:** We added "the analysis of the deterministic run is computed by applying the Kalman gain for the ensemble mean to the innovation of the deterministic run (Schraff et al. 2016)".

5. l148-149, is this the opposite? The correlations between integrated mass flux divergence and surface pressure tendency are mainly "inside" the convective regions?

**Answer:** The integrated mass-flux divergence is proportional (in magnitude) to the surface pressure tendency in case of hydrostatic pressure (i.e., non-convective regions) as shown by Eq. (2). It can be clearly seen in the last column of Fig. 3 that the patterns of integrated mass-flux divergence and the surface pressure tendency are comparable in the non-convective regions, and the former one has much more strong signals within the convective regions. We rephrased the sentence.

6. l157-159, it would be helpful to provide some explanations for the degradation of errors caused by the integrated mass flux divergence. Intuitively, by adjusting the integrated mass flux, a more balanced analysis could be obtained, which is preferable for improved forecasts. Could this intuitive hypothesis be true for large scale applications? Since E_VrZ_6m_f has larger errors and spread than E_VrZ_6m, especially for later times, E_VrZ_6m_f might have smaller increments than E_VrZ_6m. The smaller increments might not be large enough to correct the prior errors, although it is better balanced?

**Answer:** It should be emphasized that the integrated mass-flux divergence filter is a post-processing method. The analysis is an "optimal" estimate in terms of RMSE of the model state, however, it may be poorly balanced if for instance too small localization radius is applied or observations are very unevenly distributed. The divergence filter is done after the analysis step, it is aimed to reduce the imbalance but does not take the accuracy of model state into account, thus the RMSE of model may increase after this post-processing.

7. l175, why the spin-up shows in E_VrZ_6m_f but not in E_VrZ_6m? Since the spinup show in E_VrZ_6m, does it mean that the adjustment based on integrated mass flux is too much?

**Answer:** The filter may introduce some unbalanced modes that are not the solutions of the governing equations of the model. This can be attributed to the fact that the filter currently depends on the distribution function $f$, which is defined in an ad hoc manner. As stated in outlook, we are going to extend the filter to 3D, and the wind is corrected by analyzed mass-flux from level to level. This may lead to a more balanced filter.

8. l178-86, the results of forecasting are based one forecast launched at 15:00 UTC. To draw statistically significant conclusions, more than one forecast is preferred. Can the forecast be launched at different hours, so that more general conclusions can be obtained?

**Answer:** Statistical significance has been rarely used in the discussion of OSSE results for convective-scale data assimilation (e.g., Snyder and Zhang, 2003; Tong and Xue, 2004) since the test period is very short and the amount of samples is very small. To obtain somehow more robust statistics for the given circumstances, ensemble

forecasts are used in addition to the deterministic run. We emphasized this in the text.